# Zebrafish (*Danio rerio*) Prefer Undisturbed Shoals over Shoals Exposed to the Synthetic Alarm Substance Hypoxanthine-3N-oxide (C_5_H_4_N_4_O_2_)

**DOI:** 10.3390/biology14030233

**Published:** 2025-02-25

**Authors:** Andrew Velkey, Kaitlyn Kinslow, Megan Bowers, Ethan Hoffman, Jamie Martin, Bandhavi Surisetty

**Affiliations:** 1Neuroscience Program, Christopher Newport University, Newport News, VA 23607, USA; kaitlyn.kinslow.23@cnu.edu (K.K.); cara.martin.21@cnu.edu (J.M.); 2Department of Molecular Biology & Chemistry, Christopher Newport University, Newport News, VA 23607, USA; megan.bowers.21@cnu.edu (M.B.); ethan.hoffman.20@cnu.edu (E.H.); bandhavi.surisetty.20@cnu.edu (B.S.)

**Keywords:** alarm substance, predation threat, 3-chamber open-tank free-swim task, shoaling, social preference

## Abstract

This study investigates how zebrafish, a small freshwater fish known for social behavior, respond to visual indicators of potential threats in their environment. Animals often form groups to reduce the risk of predation, benefiting from safety in numbers. Zebrafish are highly social and can communicate threats through chemical signals; however, these signals are limited to short distances and can break down quickly in water. This experiment explored whether zebrafish can use visual cues alone to distinguish between groups of fish which are alarmed (indicating a potential threat) and fish which are not. Using a 3-tank experimental setup, researchers found that zebrafish preferred to stay near non-alarmed groups, spending more time and exhibiting calmer behavior in these zones. Males showed more freezing behavior than females, indicating differences in how each sex responds to threats. These findings highlight zebrafish’s ability to visually assess danger and choose safer social environments. This research has broader implications for understanding group behavior and decision-making under threat, offering insights into how animals, including humans, respond to social and environmental cues. By studying zebrafish, scientists can uncover mechanisms that may help address social- and anxiety-related disorders in humans.

## 1. Introduction

Anti-predation adaptations have evolved through natural selection to reduce risk among members of prey species and increase survival/fitness. Many animals exhibit behavioral strategies—such as playing dead or evading capture—to reduce predation risk [1,2,3]. Another common behavioral adaptation is social collection; prey often form complex social groups, which provide multiple advantages in reducing predation risk [4]. First, while predators can locate groups of prey more readily than solitary prey and can proficiently identify which prey in a group are easier targets [5,6,7,8], group living decreases the likelihood of any one individual being successfully captured due to the dilution effect [9,10], the “many eyes” effect [11,12], and the confusion effect [13,14]. Additionally, a group of vigilant individuals is more likely to detect a potential predatory threat than a solitary individual [15,16]. Once detected, threats can be communicated to other group members in a number of different ways, including alarm movements (e.g., [17,18]) and chemical messaging (e.g., [19,20]). While chemosignaling of a predatory threat can be highly specific [21,22], this signaling process requires the receiver’s proximity to the signaling animal. Chemical signals can break down quickly in aquatic environments [23,24,25], and the dynamics of concentration changes as chemical signals disperse can be quite complex [26]. It would also be adaptive for an aquatic species such as zebrafish to communicate a potential predatory threat at greater distances through visual display rather than relying solely on short-range chemosignaling. The goal of the present study is to determine whether zebrafish can use visual-only cues to detect the communication of simulated predatory threat, and if so, what social preference zebrafish will form in response to the visual detection of the communicated threat.

Zebrafish (*Danio rerio*) are highly social freshwater fish widely recognized as valuable models for biomedical and behavioral neuroscience research [27,28]. This species is particularly advantageous as a model organism due to its genetic similarity to humans [29], standardization of husbandry (e.g., [30]), rapid reproduction [31], and distinctly observable social behaviors [32]. Zebrafish exhibit a broad range of complex social interactions, many of which parallel human social behaviors, making zebrafish a powerful tool for studying neuropsychiatric disorders such as anxiety (e.g., [33]), depression (e.g., [34]), and social impairments (e.g., [35]). Their use in research enables scientists to investigate behaviors and responses to social cues in a controlled setting without the ethical and practical challenges involved in using human subjects (e.g., [36]). A key component of zebrafish social behavior is shoaling—the formation of loosely coordinated groups composed of a few to several hundred individuals [15,37]. Shoaling provides important survival benefits, such as increased mating opportunities, improved foraging efficiency, and added protection from predators through the safety-in-numbers effect [38,39,40]. These social behaviors, combined with their genetic tractability, make zebrafish an ideal model for understanding the neural and behavioral mechanisms underlying social dynamics, offering valuable insights into human health and disease.

Schreckstoff, which translates to “scary substance” in German, is a chemical message released by an injured fish that serves as a warning to nearby conspecifics of a potential predatory threat [41,42]. When epidermal injury results from a predatory attack, chemical compounds are released from specialized epidermal club cells [19,43,44] and then detected by the olfactory receptors of surrounding conspecifics [42]. Olfactory sensory neurons transmit excitatory signals to the olfactory bulb, triggering specific anxiety-like behaviors such as freezing, darting, and jumping [22,45,46,47]. Previous studies in the literature indicate that the synthetic chemical hypoxanthine-3N-oxide (H3NO) elicits alarm responses in zebrafish in a manner similar to schreckstoff [48]. Additionally, more recent research suggests that specific combinations of epidermal secretions (e.g., a fear signal plus a conspecific signal) are responsible for eliciting fear responses within a shoal [21,22]. When isolated from their home shoal and seeking new shoaling opportunities, solitary fish may avoid shoals that have experienced recent predatory attack, instead preferring to join shoals that are undisturbed [48,49].

Previous research indicates that multi-chamber tasks and multidimensional analyses enhance our understanding of zebrafish behavior [50,51,52,53]. A recent study used an adjacent-tank apparatus [54] to examine zebrafish’s responses to naturally derived alarm signals [52] in order to study social contagion—a phenomena in which undisturbed subjects respond to conspecifics exposed to an alarm signal. In this study, both undisturbed male and female subjects demonstrated freezing responses when exposed to the sight of alarmed conspecifics, with males demonstrating more freezing than females. Subjects of both sexes also demonstrated other responses to the sight of alarmed conspecifics. While the adjacent-tank configuration is useful for testing the detection of visual-only cues between subjects and stimulus shoals, it is limited in the extent to which social preference can be studied as only one social stimulus is presented to the solitary subject. More recently, the 3-chamber open-tank free-swim task (OTFST) emerged as a validated method for testing social preference in zebrafish as it increases the amount of potential data which can be obtained and allows dichotomous preference analyses [55]. Subsequent research examined zebrafish’s social preference between two different shoals of fish using the 3-chamber OTFST, demonstrating that zebrafish spend more time near real shoals than artificial shoals [56] and prefer established shoals over newly formed shoals [57]. These findings indicate that zebrafish can effectively detect subtle differences in neighboring shoals through visual cues during the 3-chamber OTFST. To further explore and develop the utility of this technique, the present study expands upon the abovementioned two-tank approaches in assessing the shoaling preference of solitary subjects. Unlike chemosignaling of predatory threats, there is limited research on zebrafish’s ability to detect visual movement signals associated with predatory threats, particularly when subjects rely solely upon visual signals in the context of the 3-chamber OTFST. The present experiment tests the hypothesis that zebrafish can detect differences between alarmed and unalarmed shoals and will spend more time near the intact (i.e., unalarmed) versus the alarmed shoal using the 3-chamber OTFST. These results would provide evidence to indicate that zebrafish can effectively communicate complex states—such as fear/anxiety—solely through visual means, allowing conspecifics to make potentially life-saving responses. In spending more time by the unalarmed shoal, the subjects would display a preference for avoiding potential danger rather than remaining close to it. Lingering near the alarmed shoal would bolster the aforementioned benefits of social collection, but avoidance would further remove the subject from potential danger, therefore increasing its chances of survival.

## 2. Materials and Methods

### 2.1. Animals and Housing

The experimental subjects (*N* = 20, 10 males and 10 females) and all stimulus fish were adult (>6 mos age) short-fin wild-type (SFWT) zebrafish obtained from a regional supplier (Quinn’s Fins, Palm Bay, FL, USA). Several previous studies have established the efficacy of using SFWT zebrafish [58,59,60] as SFWT fish are expected “to be more representative of the species and to possess fewer idiosyncratic features that may have developed during the inbreeding process of laboratory strains” [48] (p. 338); as such, this particular population is an excellent choice for behavioral studies. Upon arrival at the facility, fish were separated into same-sex group-housing tanks (38 L capacity, 50.8 cm × 25.4 cm × 30.5 cm) equipped with active bio-filters (“Penguin 100”, Model PF00100B, MarineLand—Spectrum Brands Pet, LLC, Blacksburg, VA, USA) and conditioned water (balanced pH of 7.0–8.0, 400–700 µS, temperature 25–27 °C, <40 ppm nitrates, <0.2 ppm nitrites, and 0.01–0.1 ammonia) with an average density of 1.5 fish per gallon (3.79 L). Fish were held in same-sex group-housing tanks until selection as either individual subjects or members of stimulus shoals. All fish were kept under a 14L:10D photoperiod and fed once daily (“TetraPro Tropical Crisps”—Product #77070, Tetra—Spectrum Brands Pet, LLC, Blacksburg, VA, USA). Animals were kept under these conditions for 1 to 6 weeks before being transferred to smaller staging tanks prior to experimentation.

Subjects and stimulus fish were selected on a weekly basis for the experiment and transferred to staging tanks on a stand-alone recirculating-flow rack system (#ZS660, Aquaneering, Inc., San Diego, CA, USA) maintained within the same parameters as the group-housing tanks. A high-density polyethylene tank (90 L capacity, NorthStar #2691, Northern Tool & Equipment, Burnsville, MN, USA) was custom-fitted with a flow-through manifold on the rear of the rack system to supply water for filling testing tanks. Subjects were housed individually in 1.4 L tanks (#ZT080, Aquaneering, Inc., San Diego, CA, USA) on one row of the rack system with opaque dividers placed between the tanks. Other fish were selected to form small stimulus shoals of four fish (2 males and 2 females); shoals were housed in 2.8 L tanks (#ZT280, Aquaneering, Inc., San Diego, CA, USA) on different rows of the rack system. Subjects and stimulus fish were kept under these conditions for three to seven days before testing. Following transfer of subjects and stimulus fish to the testing apparatus, all 1.4 L and 2.8 L tanks were washed and sterilized in a standard laboratory washer before being returned to the rack for subsequent re-use. s. All housing, caretaking, and other procedures involving the animals were performed under appropriate animal welfare guidelines [61].

### 2.2. H3NO Solution Preparation

An amount of 100 mg of H3NO (152.11 g/mol) was synthesized to 99% purity by a scientific laboratory (Synchem UG, Altenburg, Germany). A 50 µM stock solution was prepared in a 1 L volumetric flask by adding 7.6055 mg H3NO to reagent-grade RO water (>18 MΩ) and stored at 1 °C under dark conditions. A 5 µM working solution was prepared by adding 100 mL of stock solution to 900 mL reagent-grade RO water and also stored at 1 °C under dark conditions. Weekly, 5 mL aliquots were taken from the working solution and stored at −20 °C under dark conditions until needed for experimental sessions.

### 2.3. Apparatus and Software

Previous studies have successfully demonstrated the efficacy of the 3-chamber OTFST and include detailed descriptions of the apparatus [55,56,57]. The 3-chamber OTSFT apparatus in the present study consisted of a central testing tank (“Rimless 5.5 Gallon”—#100541424, Aqueon, Inc., Franklin, WI, USA—dimensions 41.275 cm L × 21.273 cm W × 26.67 cm H; max capacity 23.4 L) flanked by two stimulus tanks identical to the central tank (See Figure 1). This particular tank size was selected to ensure that zebrafish subjects as well as shoal mates could freely swim within their respective enclosures while still maintaining the subjects’ ability to visually assess cues from both flanking shoal stimuli without requiring excessive travel distance, which can interfere with the preference test [62]. Linear polarizing filters (dimensions, Rosco #7300, obtained from B&H Foto & Electronics, Inc., New York, NY, USA) were placed in a 90° orientation to each other on the outer surfaces of the central tank to ensure the fish of one stimulus shoal tank could not see the members of the stimulus shoal in the opposite flanking tank while still allowing the test subject in the central tank a view of both stimulus shoals. The central tank provided a free-swim environment for observing the behavioral responses of test subjects to the shoaling stimuli contained in either flanking tank. Each tank was fitted with a custom clear polycarbonate lid (21 cm × 41 cm), and the lid for the rightmost tank had a 5 mm diameter hole drilled in the center to allow for the insertion of a syringe-like Positive-Displacement Dispenser Tip (PDDT; 10 mL “Combitip” #0030089464, Eppendorf Corp., Hamburg, Germany).

Uniform backlighting for the 3-chamber OTFST apparatus was provided by an LED dimmable flat-panel light (model—CPANL 1 × 4 40LM SWW7 120 TD DCMK, Lithonia Lighting—Acuity Brands, Inc., Conyers, GA, USA) held in a custom PVC frame and controlled with a dimmer switch (model CTCL-153P-WH, Lutron Electronics, Inc., Coopersburg, PA, USA) to maintain 350 Lux at surface of the light. The rear walls of the testing chamber and flanking tanks were covered with a self-adhesive white plastic film (#16F-C9A952-06, Con-Tact Brand, La Mirada, CA, USA) which further diffused the backlighting, reducing the brightness to 15 lux at the front of each tank. One digital video camera (model—acA 1300-60gc, Basler AG, Ahrensburg, Germany) was clamped to a custom stationary frame positioned 66 cm in front of the center tank of the 3-chamber OTFST apparatus, and the camera was positioned such that the image of the entire front of the tank filled the frame. Two additional digital video cameras were clamped to the same stationary frame and positioned such that the image of each flanking tank filled the frame for its respective camera. The cameras were connected via a CAT-5 ethernet cable to a PC-compatible computer (Dell Precision 3630, Dell Computer, Inc., Round Rock, TX, USA) running Microsoft Windows 10. During experimental sessions, video from the cameras was obtained using MediaRecorder 6.0 (Noldus, Inc., Wageningen, The Netherlands). For quantification of movement variables, digital videos were analyzed using Ethovision XT 16.0 (Noldus, Inc., Wageningen, The Netherlands). The entire area for the testing apparatus and camera frame was surrounded by three custom-built floor-standing screens covered in black cloth to prevent animals on the testing bench from seeing the experimenters, and the computer was contained on a rack outside of the screened area.

### 2.4. Experimental Procedure

At the beginning of the testing day, opaque barriers attached to strings on a pulley system were placed between the testing tanks. To maintain identical water quality parameters from the staging tanks to the testing tanks and reduce transfer stress on animals [63], water was obtained from the reservoir on the flow-through rack system and used to fill the testing tanks at the beginning of each experimental day. Because net capture and air transfer can act as stressors and negatively affect the subsequent behavior of zebrafish [64], each subject and its associated stimulus shoals were volume-transferred by carefully pouring the entire contents of each staging tank into its respective tank on the 3-chamber OTFST apparatus at 09:00 on the testing day. After the transfer of staging tank contents, each tank on the testing apparatus had a final volume of 26.7 L of water. Because no main effects or interactions of side position have been revealed in previous studies by our laboratory [56,57], and counterbalancing stimulus shoal positions across the flanking tanks would result in unnecessary duplication [61] by doubling the number of animals needed for the experiment, the positions of the intact and alarmed shoals were not counterbalanced. The intact shoal was always placed in the left flanking tank, while the shoal to be alarmed was always placed in the right flanking tank. Following addition of the test subject and stimulus shoals to their respective tanks, an aerator was placed in each tank to maintain oxygen levels during the day, and a clear lid was placed on each tank. One 5 mL aliquot of the H3NO working solution was removed from the freezer and thawed to room temperature (22–24 °C). Subjects and stimulus shoals were then acclimated for 4 h before testing at 13:00.

Before starting each experimental session, a welfare check was conducted to ensure that all animals in the apparatus were responding normally. This included observing whether the animals had acclimated properly to the tanks and displayed normal behavioral patterns. Specific attention was given to identifying any signs of immobility, ensuring that shoaling behavior was present among the animals in the flanking tanks, and confirming the absence of any abnormal swimming patterns or physical abnormalities. These checks ensured the validity of the experiment by minimizing potential variability caused by stress or compromised animal health. Following the welfare check, aerators were removed from each testing tank and lids replaced. Because H3NO degrades quickly in acidic conditions [48,65], the pH was obtained for the water in the right flanking tank and tested with a bench meter to ensure it was 7.0 or higher prior to starting the experimental session. A PDDT was filled with the contents of the aliquot. The experimenter opened a session in MediaRecorder and began recording as the dividers between the testing tanks were removed. Within 30 s, the experimenter delivered the H3NO by reaching from behind the screen to fully insert the PDDT through the hole in the lid, quickly dispensing the contents into the tank to provide a final concentration of 1.5 nM of H3NO. This 1.5 nM concentration has been demonstrated to activate detectable alarm responses in zebrafish, including erratic movements [48]. Recording continued for 10 min to obtain digital video of each subject’s movement throughout the session. At the conclusion of the session, the test subject and stimulus shoal exposed to H3NO were transferred to an outbound holding tank. The intact stimulus shoal was transferred to a 2.8 L staging tank and placed on another row of the flow-through rack system to be used as an alarmed shoal in a future session at least one week later. This step allowed for further reduction in the total number of animals necessary to run the experiment [66,67,68]. After all animals were removed from the testing tanks, the tanks were drained, rinsed thoroughly, and sprayed with a 5% citric acid solution to sanitize the tanks and degrade any residual H3NO. Tanks were allowed to air dry overnight before being rinsed again at the beginning of the subsequent experimental day and returned to the testing bench.

At the conclusion of all testing sessions, the digital video from subjects’ sessions was analyzed in Ethovision XT 16.0. The arena settings for Ethovision XT 16.0 were set to divide the screen into the leftmost third of the arena (closest to the intact shoal), the middle third of the arena, and the rightmost third of the arena (closest to the alarmed shoal). These zones were used for subsequent data extraction and movement analyses.

### 2.5. Design and Analysis of Subjects’ Responses During 3-Chamber OTFST

The experiment was a 2 (sex: male or female—between subjects) × 3 (vertical zone: intact shoal third, middle third, and alarmed shoal third—within subjects) mixed-factorial design. The following measures were obtained using Ethovision XT 16.0 to analyze the digital videos from subjects’ sessions:Percent duration within each zone: The relative amount of total time each subject spent in each zone during the entire session.Percent of session time spent moving: The relative amount of time each subject was moving at any velocity in each zone.Percent of session time spent freezing: The relative amount of time each subject had ceased any detectable movement for a minimum of 3 s in each zone.Average movement velocity: The average movement speed in mm/s for each subject’s movement within each zone.

Data were organized into a 2 (sex) × 3 (vertical zone) design and analyzed with SPSS 29.01 (IBM Corp., Armonk, NY, USA) using a Linear Mixed Model (LMM) with Type III Sums of Squares at α = 0.05; for advantages of the LMM over mixed-factorial ANOVA, see [69]. In addition to the advantages of the LMM over mixed-factorial ANOVA, the LMM is extremely robust when the normality assumption is violated [70]. Exploration of the current data revealed multiple distributions which did not conform to a normal distribution and demonstrated heterogeneity of variance. While there may be concerns with deploying the LMM when the normality assumptions are not met, recent work indicates that alternative approaches, such as attempting to normalize the data via transformation, may be more problematic [71]. Accordingly, the model was set with diagonal covariance structure for heterogeneous variance. Degrees of freedom for the denominator of mixed-model F-ratios were adjusted according to the Maximum Likelihood Estimator for LMM. When appropriate, unplanned comparisons were made using the Bonferroni correction for family-wise error. Experimental effect sizes are reported as partial-η^2^ values (obtained from SPSS 29.01) as well as Cohen’s *f* values (obtained using an on-line calculator [72]). LMM fit is reported as *R*^2^ values (obtained using an on-line calculator [73]). LMM performance summaries are provided as Appendix A.

## 3. Results

### 3.1. Manipulation Check and Qualitative Findings

To verify that H3NO exposure had a demonstrable effect on the activity of shoal members, digital video footage of shoal responses was analyzed with Ethovision 16.0 to obtain the mean inter-individual distances (IIDs) for each stimulus shoal during each session. These data were organized into a between-subjects design (alarmed shoal vs. intact shoal) and analyzed using an independent-samples *t*-test with SPSS 29.01 at α = 0.05. The mean IID during each session was calculated for both types of shoals. As Levene’s test for equality of variance was not significant (*F*(1, 38) = 3.37, *p* = 0.07), equality of variances was assumed for the subsequent independent-samples *t*-test. The mean IID for members of intact shoals (*M* = 10.42 cm, *SEM* = 0.56) was significantly greater than the mean IID for members of alarmed shoals (*M* = 8.67 cm, *SEM* = 0.83) (*t*(38) = 1.75, *p* = 0.044, Cohen’s *d* = 0.55, η_p_^2^ = 0.081). The effect size on IID due to the administration of the synthetic alarm substance is considered “medium” under Cohen’s criteria [74] p. 26 and accounts for 8.1% of the variance in IID across sessions (see Figure 2). Alarmed shoals had a lower mean IID than intact shoals, which is consistent with previous reports (e.g., [51,75]).

Ethovision was used to generate a location heatmap by aggregating movement tracks across subjects. Based on this heatmap, subjects spent more time in the zone adjacent to the unalarmed shoal compared to the time spent in the zone adjacent to the alarmed shoal, as depicted in the location heat map generated with Ethovision 16.0 (see Figure 3). This result indicates that subjects were able to distinguish between alarmed and intact shoals and demonstrated a preference for the intact shoal. Due to the qualitative nature of this finding, statistical analyses were performed across the various measures in order to further quantify the nature of this preference.

### 3.2. Percent of Session Time in Zones

The analysis of subjects’ percent duration in the different zones revealed a significant main effect for zone (*F*(2, 32.25) = 43.5, *p* ≤ 0.001, *R*^2^ = 0.706, η_p_^2^ = 0.73, Cohen’s *f* = 1.55); the effect size is considered “ large” under Cohen’s criteria [74] p. 287 and accounts for 73% of the variance in percent duration in different zones. However, there was neither a significant main effect for sex nor an interaction of sex by zone (both *F*s ≤ 0.2, *p*’s ≥ 0.682). On average, subjects spent almost twice as much time in the zone closer to the intact shoal (*M* = 64.7%, *SEM* = 7.24%, *SD* = 32.77%) than they spent in the zone closer to the alarmed shoal (*M* = 34.67%, *SEM* = 7.46%, *SD* = 34.25%). Subjects spent the least amount of time in the middle zone of the arena (*M* = 3.56%, *SEM* = 0.75%, *SD* = 3.37%; see Figure 4).

### 3.3. Percent of Time in Motion

The analysis of subjects’ percent of session time in motion in the different zones revealed a significant effect for zone (*F*(2, 32.35) = 31.84, *p* ≤ 0.001, *R*^2^ = 0.635, η_p_^2^ = 0.66, *f* = 1.32); the effect size is considered “large” under Cohen’s criteria [74] p. 287 and accounts for 66% of the variance in percent time in motion across the different zones. However, there was neither a significant main effect for sex nor an interaction of sex by zone (both *F*s ≤ 0.14, *p*’s ≥ 0.136). Subjects spent a greater percentage of time moving in zones closer to either the intact shoal (*M* = 46.97%, *SEM* = 6.04%, *SD* = 27.02%) or the alarmed shoal (*M* = 26.06%, *SEM* = 6.23%, *SD* = 28.71%) than they did in the middle zone of the arena (*M* = 3.41%, *SEM* = 0.710, *SD* = 3.22%; see Figure 5).

To characterize the potential effects of zone and/or sex in a more relative manner, the total time in motion per zone (in S) was divided by the total time in zone (in S) and multiplied by 100 to provide a percentage of time in zone spent in motion (versus percent of overall session time spent in motion). The analysis of the subjects’ percent of time in zone spent in motion revealed a significant effect for zone (*F*(2, 30.96) = 38.55, *p* ≤ 0.001, *R*^2^ = 0.687, η^2^ = 0.71, *f* = 1.48); the effect size is considered “large” under Cohen’s criteria [74] p. 287 and accounts for 71% of the variance in percent time in motion across the different zones. However, there was neither a significant main effect for sex nor an interaction of sex by zone (both *F*s ≤ 2.38, *p*’s ≥ 0.109). Subjects spent a greater percentage of time in zone in motion zones closer to either the intact shoal (*M* = 72.6%, *SEM* = 2.84%, *SD* = 13.79%) or the alarmed shoal (*M* = 79.3%, *SEM* = 3.50%, *SD* = 16.12%) than they did in the middle zone of the arena (*M* = 96.1%, *SEM* = 1.00%, *SD* = 4.45%; see Figure 6).

### 3.4. Percent of Time Without Motion (Freezing)

The analysis of subjects’ percent of session time without motion in the different zones revealed a significant effect for zone (*F*(2, 32.42) = 49.04, *p* ≤ 0.001, *R*^2^ = 0.731, η_p_^2^ = 0.75, *f* = 1.65); the effect size is considered “large” under Cohen’s criteria [74] p. 287 and accounts for 75% of the variance in percent duration of freezing across the different zones. Regarding the main effect for zone, subjects spent more time motionless in the zone closer to the intact shoal (*M* = 17.68%, *SEM* = 1.96%, *SD* = 9.68%) than they did when closer to the alarmed shoal (*M* = 8.52%, *SEM* = 1.97%, *SD* = 9.18%). Subjects spent the least amount of time motionless in the middle zone of the arena (*M* = 0.14%, *SEM* = 0.05%, *SD* = 0.22%; see Figure 7).

The ANOVA also revealed a significant effect for sex (*F*(1, 39.01) = 4.14, *p* = 0.049, *R*^2^ = 0.073, η_p_^2^ = 0.10, *f* = 0.28); the effect size is considered “medium” under Cohen’s criteria [74] p. 286 and accounts for 10% of the variance in percent duration of freezing across the different zones. Regarding the main effect for sex, males (*M* = 10.6%, *SEM* = 1.33%, *SD* = 11.91%) spent more time motionless than females (*M* = 6.9%, *SEM* = 1.29%, *SD* = 8.53%, see Figure 8). The interaction of sex by zone on percent time freezing was not significant (*F*(2, 32.42) = 2.46, *p* = 0.101, *R*^2^ = 0.078, η_p_^2^ = 0.103, *f* = 0.29).

To characterize the potential effects of zone and/or sex in a more relative manner, the total time freezing per zone (in S) was divided by the total time in zone (in S) and multiplied by 100 to provide a percentage of time in zone spent freezing (versus percent of overall session time spent freezing). The analysis of subjects’ percentage of time in zone spent freezing revealed a significant effect for zone (*F*(2, 29.687) = 46.36, *p* ≤ 0.001, *R*^2^ = 0.736, η^2^ = 0.76, *f* = 1.67); the effect size is considered “large” under Cohen’s criteria [74] p.287 and accounts for 71% of the variance in percent time in motion across the different zones. However, there was no significant main effect for sex nor an interaction of sex by zone (both *F*s ≤ 2.08, *p*’s ≥ 0.157). Subjects spent a greater percentage of time freezing in the zone closer to either the intact shoal (*M* = 27.2%, *SEM* = 2.8%, *SD* = 13.63%) or the alarmed shoal (*M* = 20.4%, *SEM* = 3.49%, *SD* = 16.10%) than they did in the middle zone of the arena (*M* = 2.6%, *SEM* = 0.74%, *SD* = 3.34%; see Figure 9).

### 3.5. Velocity During Movement

The analysis of subjects’ movement velocity revealed a significant effect for zone (*F*(2, 23.63) = 14.28, *p* ≤ 0.001, *R*^2^ = 0.098, η_p_^2^ = 0.17, *f* = 0.33); the effect size is considered “medium” under Cohen’s criteria [74] p. 286 and accounts for 17% of the variance in velocity during movement across the different zones. However, there was neither a significant main effect for sex nor an interaction of sex by zone (both *F*s ≤ 0.83, *p*’s ≥ 0.38). Regarding the main effect of zone, subjects swam the slowest average speed in the zone closer to the intact shoal (*M* = 4.57 mm/s, *SEM* = 0.26 mm/s, *SD* = 1.21 mm/s). There was no significant difference in the average swim speed between the middle zone (*M* = 10.30 mm/s, *SEM* = 1.10 mm/s, *SD* = 4.95 mm/s) and the zone proximal to the alarmed shoal (*M* = 8.54 mm/s, *SEM* = 2.16 mm/s, *SD* = 10.22 mm/s; see Figure 10).

## 4. Discussion

In a previous experiment on social contagion and social buffering in zebrafish [52], males were responsive to visual cues indicating threat perception from alarmed conspecifics, while females demonstrated greater sensitivity to chemical alarm substances. However, the choice architecture in that study offered only an asymmetrical comparison, i.e., a “Hobson’s Choice” [76], where conspecifics were presented visually on one side of the experimental chamber, while the other side lacked any stimulus. This design provided limited insights into preference or avoidance responses as the subjects were basically choosing between the presence and the absence of conspecifics rather than evaluating among two concurrently available social stimuli. The present study addresses this limitation by using a symmetrical choice architecture, presenting zebrafish with two shoaling options: an intact shoal and an alarmed shoal. The present approach provides a more comprehensive assessment of social preference under conditions specifically designed to test for social cueing of fear responses and predatory threats.

Our findings are consistent with prior research [52], demonstrating that both male and female zebrafish in the present study demonstrate social contagion of fear when visually exposed to alarmed conspecifics. Specifically, subjects of both sexes exhibited a clear preference for proximity to the intact shoal over the alarmed shoal, spending significantly more time in zones closer to the intact shoal. These results suggest that zebrafish can effectively differentiate between alarmed and unalarmed groups solely by relying on visual cues to assess threat levels. By preferentially associating with less-alarmed conspecifics, zebrafish can mitigate predation risk while maintaining the benefits of social grouping [9,13,15]. These behaviors highlight the utility of zebrafish as a model organism for studying complex social processes in response to environmental threats.

Interestingly, our study also revealed a significant sex difference in freezing behavior as a percentage of the total session time, with males freezing more than females. This finding aligns with previous research [52] which demonstrated that males are more responsive to visual information related to the predatory threat. This sex-specific behavior may reflect differences in life history strategies, where males adopt higher-risk approaches that prioritize quick reactions to visual cues, even at the expense of greater energy expenditure [77,78]. In contrast, females may rely more heavily on chemical alarm cues, which are slower to disseminate but may provide more reliable information about the presence of a predator. These sex differences underscore the importance of considering individual differences in threat perception and response when interpreting zebrafish behavior.

The current study has several limitations that could be addressed in future research to enhance the robustness of the findings. As all subjects and stimulus fish were obtained as adults from an external source, the variability in the exact age of the animals likely increased individual variations, as behavioral responses such as freezing and velocity of movement may vary with developmental or age-related changes in social behavior or sensory processing. Additionally, the absence of developmental history data—including prior social exposure or environmental conditions—further limits the interpretation of the results as these factors could influence both social preferences and threat perception. A future study could utilize animals raised within a facility to provide enhanced controls across the lifespan of the subjects and offer precisely specified ages for subjects and stimulus animals. Third, the lack of a negative control, where neither stimulus shoal receives schreckstoff, limited our ability to collect measures on baseline activity rates and compare normal versus experimental-induced behaviors. Obtaining baseline activity measures in the future would provide additional within-subject measures to better address likely individual differences in motility and movement parameters. Finally, the manual method for adding the alarm substance to the stimulus tank may have introduced variability due to personal differences in how each experimenter specifically delivered the schreckstoff solution; this problem could be mitigated in future studies through the use of a standardized, automated delivery system (e.g., a remotely operated syringe pump) to ensure consistent dosing across trials with minimal experimenter presence. Addressing these limitations in the future would provide more controlled and reliable insights into zebrafish’s social and threat behaviors.

As SFWT fish were used as subjects in the current study rather than a more constrained laboratory strain, the findings of this study are further strengthened by their ecological validity and potential relevance to wild zebrafish populations. In natural environments, zebrafish are exposed to predatory threats, and their ability to visually discern between intact and alarmed shoals is likely critical for survival. The current study provides a contrived instance of a potentially real scenario: in nature, fish can become separated from their shoal for a variety of reasons. The capability to identify a potential predatory threat at a distance using visual information may provide the lone animal critical time to flee a shoal which has been recently attacked and affiliate instead with an intact shoal. Furthermore, fish and other wild animals are likely to reduce foraging activity in areas where predatory attacks occur with more frequency (e.g., [79]). While these animals may not abandon these areas entirely, they will reduce their foraging in predator-prone areas, even if they select sub-optimal patches to do so. A future study could potentially assess the relationship between predator avoidance and foraging approach to explore the dynamics of these competing interests.

Using the 3-chamber OTFST, there are multiple avenues for future research. While the present study investigated one dose of schreckstoff in testing the final 1.5 nM concentration of H3NO on the alarmed shoal, a future study could use the current research paradigm to investigate several doses of schreckstoff, potentially including doses previously tested by others [48], to characterize a behavioral dose–response curve. Higher doses would be expected to elicit more pronounced alarm responses, providing insights into whether zebrafish subjects exhibit a stronger preference for unalarmed shoals as the alarm intensity increases. Such a finding would further indicate the strength and sensitivity of zebrafish to visual-only cues regarding anti-predation responses. Alternatively, a developmental approach could explore these effects in larval zebrafish. Recent research indicates that days-old zebrafish larvae rapidly learn predator detection within their first week [80]; conducting a similar 3-chamber OTFST experiment with tanks appropriately sized for larval zebrafish could clarify whether these social and anti-predatory behaviors emerge early or develop later. Such research would further bridge the gap between neural mechanisms of development and behavioral responses, providing a developmental link to adult behaviors. Finally, future studies could examine the interaction of schreckstoff-induced alarm behaviors with anxiolytic substances, such as benzodiazepines [81], which may dampen alarm signaling, or anxiogenic substances such as caffeine [82], which may exacerbate shoaling deterioration due to heightened anxiety and anti-predation responding. These pharmacological tests could be performed on subjects and/or shoal stimuli. Such a multi-faceted approach could explore the interaction of alarm substances and pharmacological agents to enhance our understanding of the neural mechanisms underlying zebrafish responses as these responses pertain to social and survival behaviors.

The present findings not only contribute to our understanding of zebrafish social behavior but also offer valuable insights into the dynamics of social contagion in the context of threat perception. Future research should investigate symmetrical choice dynamics under conditions that combine social buffering and social contagion to explore potential interactions between these two phenomena. Additionally, exploring dose-dependent effects of alarm substance exposure could provide deeper insights into how varying levels of threat cues modulate zebrafish behavior. Such studies could establish the thresholds at which zebrafish shift from avoidance to engagement behaviors and enhance our understanding of the neural mechanisms underlying these responses. Much like zebrafish, humans tend toward the formation of complex social groupings for survival, security, and social connection [83]. Analyzing the behaviors of zebrafish can therefore offer insights into how humans may similarly respond to social cues. The present study indicates that zebrafish, when visually presented with an alarmed and an unalarmed shoal, prefer spending time with the unalarmed group, displaying avoidance behaviors in response to the potential threat indicated by the alarmed shoal. This behavior can be equated to the “fight or flight” responses triggered by the sympathetic activation of the autonomic nervous system in many animals, including humans, specifically the “flight” response that drives animals to flee/avoid potential danger. This area of research holds promise for advancing our knowledge of adaptive group behaviors and their implications for social responding in zebrafish, humans, and other social species.

## 5. Conclusions

Previous research on social contagion and buffering in zebrafish [52] revealed that males respond strongly to visual threat cues from alarmed conspecifics when choosing to either affiliate with or avoid a shoal, whereas females appear to rely more on chemical alarm substances when choosing to either affiliate with or avoid a shoal. By adding a second shoaling option, the present study employed a symmetrical choice architecture which offered zebrafish subjects the option to associate with either an intact shoal or an alarmed shoal, thereby enabling a clearer evaluation of social preferences under predatory threat. Consistent with prior findings, both sexes preferred proximity to the intact shoal, highlighting zebrafish’s ability to differentiate threat levels via visual cues and mitigate predation risk by associating with less-alarmed conspecifics. Notably, males exhibited more freezing behavior than females, reflecting sex-specific differences in threat perception: males may prioritize rapid visual responses, and females may rely on slower, but perhaps more reliable, chemical cues. These results underscore zebrafish’s utility as a model for studying social- and threat-related behaviors and suggest directions for future research, including the interplay between social buffering and contagion, dose-dependent alarm cue effects, and the neural mechanisms underlying adaptive group behavior.

## Figures and Tables

**Figure 1 biology-14-00233-f001:**
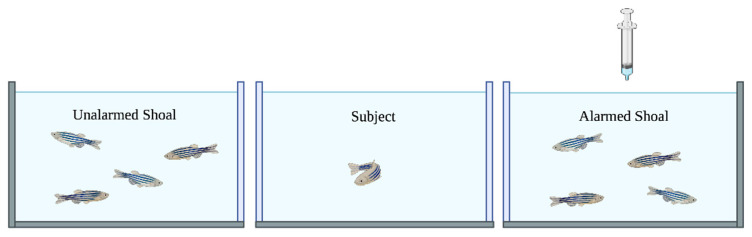
A cartoon illustration of the apparatus depicting the experimental setup. Digital cameras were placed in front of the test tank to capture digital video of the subjects’ movements during their 10 min experimental sessions. The syringe was filled with 5 mL of working solution, and the contents were manually delivered to provide a final concentration of 1.5 nM in the right tank.

**Figure 2 biology-14-00233-f002:**
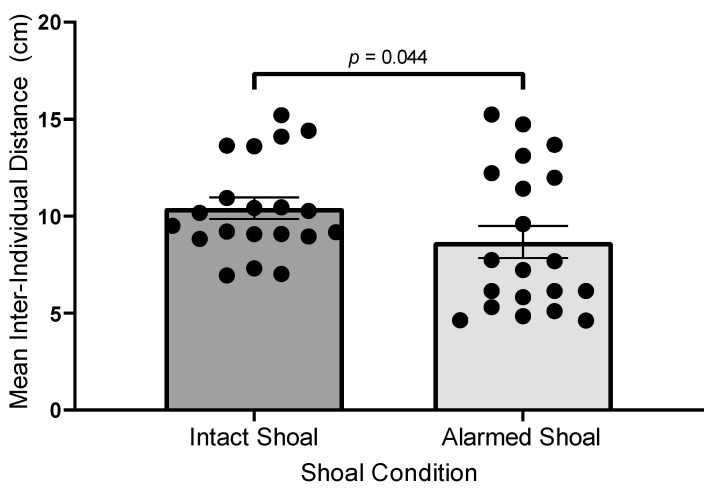
Dot plot for comparison of mean inter-individual distances (IIDs) between alarmed shoals and intact shoals. Error bars represent ±1 SEM.

**Figure 3 biology-14-00233-f003:**
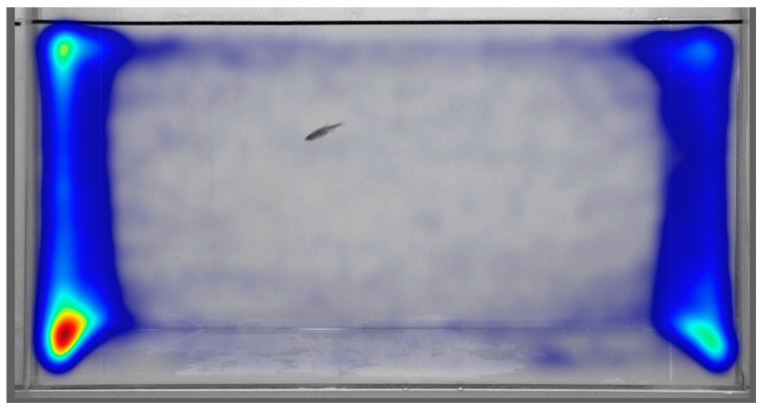
An Ethovision heatmap visualization of the subjects’ positions in the center arena. The color intensity ranging from cold (gray) to hot (red) indicates the relative frequency of sampling tracks per location; cool colors (gray to blue) indicate relatively fewer samplings of subjects in that location, while warm to hot colors (yellow/orange to red) indicate relatively more samplings of subjects in that location.

**Figure 4 biology-14-00233-f004:**
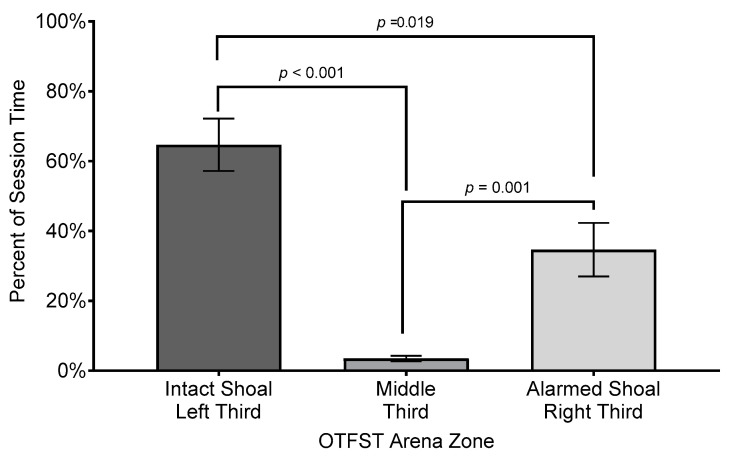
Comparison of mean percent session time subjects spent in each zone of testing arena. Difference bars between zones are reported using Bonferroni adjustment for family-wise error. Error bars represent ±1 SEM.

**Figure 5 biology-14-00233-f005:**
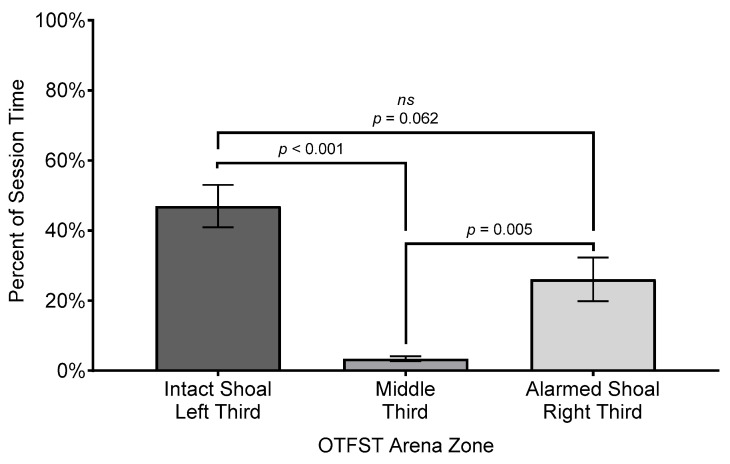
Comparison of mean percent of session time subjects spent in motion in each zone of testing arena. Difference bars between zones are reported using Bonferroni adjustment for family-wise error. Error bars represent ±1 SEM.

**Figure 6 biology-14-00233-f006:**
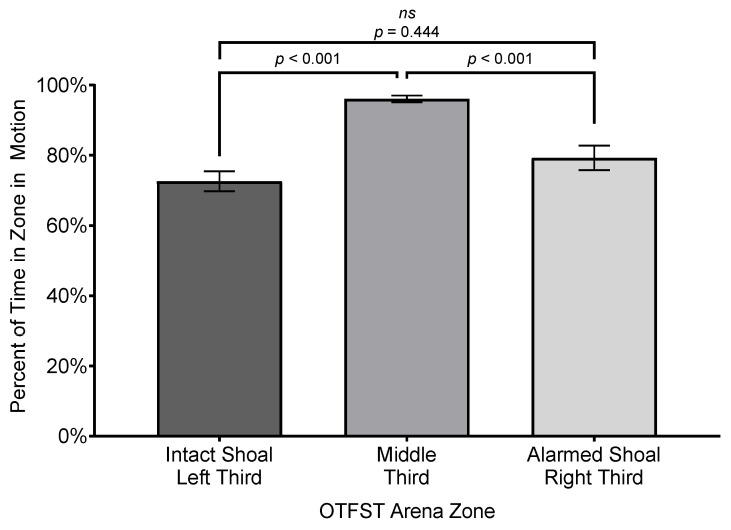
Comparison of mean percent of time subjects spent in motion in zone. Difference bars between zones are reported using Bonferroni adjustment for family-wise error. Error bars represent ±1 SEM.

**Figure 7 biology-14-00233-f007:**
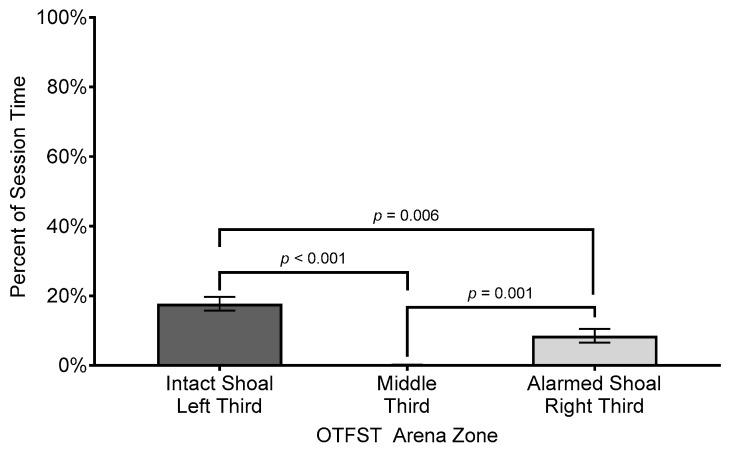
Comparison of mean percent session time subjects spent freezing in each zone of testing arena. Difference bars between zones are reported using Bonferroni adjustment for family-wise error. Error bars represent ±1 SEM.

**Figure 8 biology-14-00233-f008:**
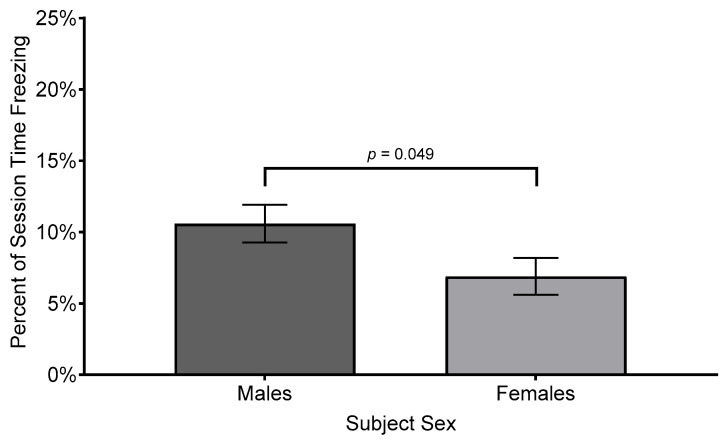
Comparison of mean percent session times male and females spent freezing in each zone of testing arena. Difference bars between zones are reported using Bonferroni adjustment for family-wise error. Error bars represent ±1 SEM.

**Figure 9 biology-14-00233-f009:**
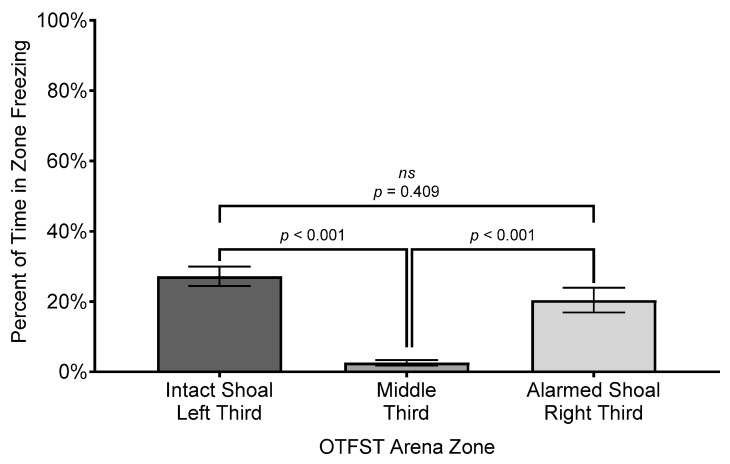
Comparison of mean percent time subjects spent freezing in zone. Difference bars between zones are reported using Bonferroni adjustment for family-wise error. Error bars represent ±1 SEM.

**Figure 10 biology-14-00233-f010:**
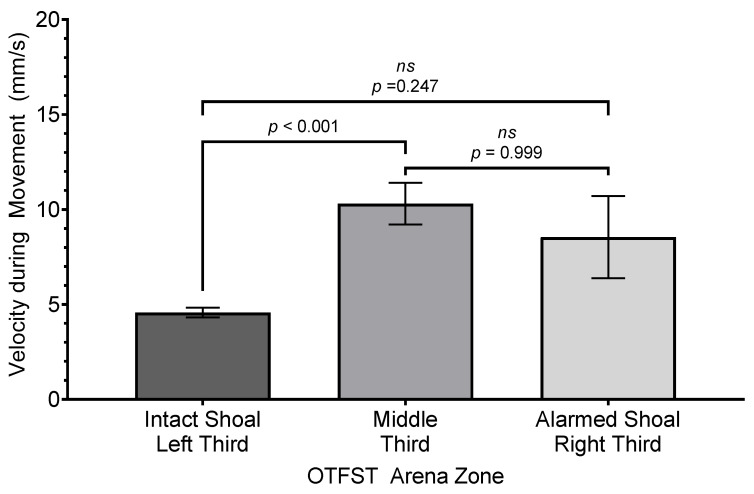
Comparison of mean velocity during subjects’ movements in each zone of testing arena. Difference bars between zones are reported using Bonferroni adjustment for family-wise error. Error bars represent ±1 SEM.

## Data Availability

The data presented in this study are available upon request from the corresponding author.

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
