# Peer review of "Zebrafish (Danio rerio) Prefer Undisturbed Shoals over Shoals Exposed to the Synthetic Alarm Substance Hypoxanthine-3N-oxide (C5H4N4O2)"

_biology, 2025, doi:10.3390/biology14030233_

Round 1

Reviewer 1 Report

Comments and Suggestions for Authors

In the manuscript by Velkey et.al., the authors have used the 3-Chamber OTFST apparatus and performed a well designed experiment to understand key aspects of zebrafish social behaviour. Their results show that zebrafish prefer intact shoals over alarmed shoals of conspecifics. Overall, the findings of this work are interesting and deserve future exploration, however, I had the following comments/suggestions.

1. A daigram of the experimental setup would be very useful in understanding the experiment.

2. Line 283. What is the size of tank used for this experiment? As larger tanks allow a more natural behaviour, I was wondering if the alarmed and the intact shoals would show a more significant difference if larger tanks could be used. If possible can the experiment be done using larger tanks.

3. Line 329. Figure 3 . Though there is a significant difference in the percent of session time in motion in between zones, it would be worth looking at what proportion of total time in each zone is in motion and is this different between zones. If it is not different between zones, based on results shown in Figure2, one can argue that since the subject stay longer near intact shoals they have higher percent of session time in motion in that zone.  

3. Line 351. Figure 4. Though there is a significant difference in the percent of session time without motion in between zones, it would be worth looking at what proportion of total time in each zone is without motion and is this different between zones. If it is not different between zones, based on results shown in Figure2, one can argue that since the subject stay longer near intact shoals they have higher percent of session time without motion in that zone. 

Reviewer 2 Report

Comments and Suggestions for Authors

Notes for authors

  1. Introduction:

    • While the introduction discusses the relevance of zebrafish as a model organism, it could benefit from a clearer articulation of the study's hypothesis and its broader implications in behavioral ecology.

  2. Literature Review:

    • The paper should better contextualize its findings within existing literature on chemosignaling and visual threat detection. For instance, it could draw more comparisons with studies on other species to highlight the novelty of its approach.

  3. Experimental Design:

    • Counterbalancing Issues: The decision not to counterbalance the positions of the intact and alarmed shoals could introduce positional bias, undermining the validity of the results.

    • Sample Size: A sample size of 20 zebrafish (10 males, 10 females) may limit the statistical power to detect subtle sex-based differences.

    • The study can benefit from a control setup where the authors explore what happens when the fish are exposed to fish who are not stressed.

    • At line 231 authors mention a welfare check was conducted before commencing each session. The authors may kindly elaborate on the same. 

    • The authors may kindly describe how the data was processed, how much data was generated. How many hours of data was collected per subject .

    • The authors should address if all the assumptions of the linear mixed effect models were met by the data. From the scatter plot shown on figure1 and given the spread of the data it seems that the data may not meet assumptions of normality. 

    • The manuscript would benefit from a diagrammatic representation of the experiment setup.

    • All candidate models should be shown in a table with their respective AIC values that led the authors to select the final model.

    • The fit of the model to the data should be better represented. 

    • Given the data, the authors may explore  generalized additive mixed effects model, which will account for the small sample size, repeated measures design.

  4. Data Analysis:

    • The LMM analysis is robust, but a supplementary visual representation of data distributions (e.g., violin plots) could provide additional insights.

    • The Figure 1 

    • The study does not address potential confounding variables, such as individual variability in baseline activity levels.

    • The authors express interest in understanding the sex differences in responses of the zebra fish but the results do not address this in a meaningful manner. The p values reported are exactly 0.05 and should have been lower to prove the hypothesis beyond reasonable doubt. 

    • The Linear mixed model design used in this study is not sufficiently discussed. How did the authors explore the data, I can see a lot of outliers in the figure 1 how does the model fit the data. What were the other candidate models. How  did the authors end up choosing this particular model and how much of the dat a was explained by the model.

    • The effect size calculations must have used means and standard deviations, which is appropriate for an experimental setup. I request the authors to share the standard deviations to get a better understanding of the range of the data. 

    • Cohens effect size of 0.2 is small, 0.5 is medium effect and 0.8 and above is considered large effect.  An effect size of 0.66 is not  a “large” effect as suggested by authors in the percent of time in motion section of the results. An affect size of 0.1 is a small effect not a medium effect as suggested by the authors in the  percent of time without motion(freezing) section of the results section.

    • None of the graphs show the sex differences in responses of the zebra fish to the stimulus.

  5. Discussion:

    • The discussion does not sufficiently explore alternative explanations for the observed behaviours, such as stress responses unrelated to predatory threat.

    • Greater emphasis on the ecological validity of the findings would strengthen the argument for their relevance to wild zebrafish populations.

    • What is the future scope of this research, what could the authors have done better? How can anyone replicating this study improve it? The authors can elaborate on these points in the discussion section.

  6. Implications for Broader Research:

    • The potential translational implications for understanding anxiety and social behavior in humans, while mentioned, could be expanded with specific hypotheses or future research directions.

  7. Clarity and Conciseness:

    • The methodology section, while detailed, is overly lengthy and could be condensed by moving some descriptions to supplementary materials.

  8. Visualization:

    • Figures illustrating experimental setup and key results are present but could be improved for clarity (e.g., contrasting color schemes, annotated zones in tank diagrams).

Recommendations:

  • Expand the Sample Size: Including more subjects and potentially replicating the study with other species would increase generalizability.

  • Counterbalancing: Implementing counterbalancing would address positional bias concerns.

  •  

Addressing these points can improve the academic rigor of this manuscript.
